# Multidrug-Resistant *Streptococcus agalactiae* Strains Found in Human and Fish with High Penicillin and Cefotaxime Non-Susceptibilities

**DOI:** 10.3390/microorganisms8071055

**Published:** 2020-07-16

**Authors:** Carmen Li, Dulmini Nanayakkara Sapugahawatte, Ying Yang, Kam Tak Wong, Norman Wai Sing Lo, Margaret Ip

**Affiliations:** Department of Microbiology, Prince of Wales Hospital, The Chinese University of Hong Kong, Hong Kong Special Administrative Region (HKSAR), Hong Kong, China; 2carmen.li@cuhk.edu.hk (C.L.); 1155085653@link.cuhk.edu.hk (D.N.S.); yingyang@cuhk.edu.hk (Y.Y.); kamtakwong@cuhk.edu.hk (K.T.W.); normanlo@cuhk.edu.hk (N.W.S.L.)

**Keywords:** *Streptococcus agalactiae*, GBS, Group B streptococci, penicillin non-susceptible, cefotaxime non-susceptible, fluoroquinolone resistance, antibiotic resistance

## Abstract

Penicillin non-susceptible *Streptococcus agalactiae* (PEN-NS GBS) has been increasingly reported, with multidrug-resistant (MDR) GBS documented in Japan. Here we identified two PEN-NS GBS strains during our surveillance studies: one from a patient’s wound and the other from a tilapia. The patient’s GBS (H21) and fish GBS (F49) were serotyped and tested for antibiotic susceptibility. Whole-genome sequencing was performed to find the sequence type, antimicrobial resistance genes, and mutations in penicillin-binding proteins (PBPs) and fluoroquinolone (FQ) resistance genes. H21 and F49 belonged to ST651, serotype Ib, and ST7, serotype Ia, respectively. H21 showed PEN and cefotaxime minimum inhibitory concentrations (MICs) of 2.0 mg/L. F49 showed PEN MIC 0.5 mg/L. H21 was MDR with *ermB*, *lnuB*, *tetS*, *ant6-Ia*, *sat4a*, and *aph3-III* antimicrobial resistance genes observed. Alignment of PBPs showed the combination of PBP1B (A95D) and 2B mutations (V80A, S147A, S160A) in H21 and a novel mutation in F49 at N192S in PBP2B. Alignment of FQ-resistant determinants revealed mutation sites on gyrA, gyrB, and parC and E in H21. To our knowledge, this is the first report of GBS isolates with such high penicillin and cefotaxime MICs. This raises the concern of emergence of MDR and PEN-NS GBS in and beyond healthcare facilities.

## 1. Introduction

Group B *Streptococcus* (GBS) is a normal commensal of the genitourinary tract as well as a major pathogenic organism of invasive infections in neonates, pregnant women, non-pregnant adults including those with underlying diseases (e.g., diabetics) [1]. GBS also causes infections in animals, including bovine mastitis and streptococcosis in farmed fish, which jeopardizes farm production [2,3]. The emergence of a GBS clone with zoonotic potential was established whereby a sepsis outbreak occurred in humans through consumption of raw fish in Singapore [4].

Two groups of antibiotics, penicillins and aminopenicillins, are recommended as first-line therapy against GBS infections; macrolides (erythromycin) and lincosamide (clindamycin) represent the second-line antibiotics that are usually prescribed for those with an allergy to beta-lactams. Penicillin G (PEN) is the drug of choice and is used widely in treatment and prevention of GBS infection, such as intrapartum antibiotic prophylaxis in pregnant mothers to prevent early onset GBS neonatal disease [5]. PEN and β-lactams are also commonly used in animal farming and aquaculture for prophylactic or treatment purposes [6]. Therefore, penicillin non-susceptibility (PEN-NS) is an important concern and may necessitate alternative options in treatment guidelines.

A multidrug-resistant (MDR) isolate refers to a strain that is resistant to three or more types of antimicrobial drugs simultaneously. The emerging antimicrobial resistance in clinical settings, communities, and veterinary medicine has become a threat to public health worldwide. MDR GBS, especially fluoroquinolone (FQ) resistance, has also been observed, mainly due to efflux mechanisms or mutations in the quinolone-resistance-determining regions (QRDRs) of the genes coding for type II topoisomerase enzymes DNA gyrase (*gyrA/gyrB*) and topoisomerase IV (*parC/parE*) [7].

We have recently identified two GBS strains with PEN-NS, one from a patient (H21) during our surveillance studies on human carriage isolates and one from a tilapia fish (F49) during our food surveillance study. The aim of the current study is to characterize these two GBS strains in view of their antibiotic resistance.

## 2. Materials and Methods

### 2.1. Bacterial Strains

Human GBS strain (H21) was isolated from a wound swab of a male aged 61 years during our surveillance studies on human carriage isolates (one of 2517 single-patient isolates surveyed during 2014–2017). Animal GBS strain (F49) was recovered from a flesh sample of a tilapia among 126 tilapias that were procured from a Hong Kong wet market during our food surveillance study in 2016.

### 2.2. Ethics

Clinical data of H21 were obtained with approval from the Joint CUHK-NTEC Clinical Research Ethics Committee (ref. no.: 2017.230). The necessary biological and chemical safety approval was obtained from the university safety office.

### 2.3. DNA Extraction and Serotyping

DNA extraction was done by emulsifying two to four bacterial colonies in 200 µL lysis buffer (0.25% sodium dodecyl sulfate, 0.05 N NaOH) at 94 °C for 5 min and centrifuged at 16,000× *g* for 5 min. Supernatant was retained and stored at −20 °C until further analysis to serve as DNA template [1]. Serotyping for capsular polysaccharide antigens I–IX was performed according to a previously described protocol [8], and gel bands were visualized using Gel Doc XR+ Gel Documentation System (BioRad Laboratories, California, USA).

### 2.4. Antimicrobial Susceptibility Testing

Both H21 and F49 strains were tested for antibiotic susceptibilities of 13 antibiotics including penicillin (PEN), cefotaxime (CTX), vancomycin (VAN), erythromycin (ERY), clindamycin (CLI), gentamicin (GEN), ciprofloxacin (CIP), levofloxacin (LEV), tetracycline (TET), minocycline (MIN), doxycycline (DOX), linezolid (LNZ), and chloramphenicol (CHL) by broth microdilution according to Clinical and Laboratory Standards Institute (CLSI) with *Streptococcus pneumoniae* ATCC 49619 as control [9]. Minimum inhibitory concentration (MIC) for the antibiotics was determined by visual inspection [9]. Cefotaxime (CTX) and PEN-NS were further confirmed by E-test according to manufacturer’s protocol (Biomérieux, France) with breakpoints referenced by CLSI [9].

### 2.5. Whole-Genome Sequencing

Whole-genome sequencing was performed by first extracting bacterial DNA with Wizard Genomic DNA Purification Kit (Promega, Madison, WI, USA) followed by library preparation and sequencing using the Nextera XT Library Preparation Kit and the Nextseq 500 System, respectively, according to manufacturer’s protocol (Illumina, San Diego, CA, USA). Approximately 50× average coverage of 150 bp pair-end sequence data was generated. Genomes were assembled as previously described [10]. Briefly, FastQC was used to perform quality control of the reads prior to assembly with SPAdes assembler (v 3.5.0) [11,12]. Contigs of at least 500 bp were included for further analyses including genome annotation through Prokka (v 1.9) [13]. MLST, antibiotic resistance genes (ARGs), and virulence factors were matched to pubMLST (www.pubmlst.org/sagalactiae), ARG-ANNOT-V3 (https://www.mediterranee-infection.com/arg-annot/), and vfdB databases [14], respectively, through ABRicate software (https://github.com/tseemann/abricate). Sequence alignment of penicillin-binding proteins (PBPs) (PBP1A, 1B, 2A, 2B, and 2X) of the isolates was performed with reference to genomes GBS 2603V/R and NEM316 (Genbank Accession No.: NC_004116.1 and NC_004368, respectively). PBP sequences from a Chinese animal study [2] were also included. Sequences of FQ-resistant determinants (gyrA/B and parC/E) were also aligned to the 2 reference genomes to identify mutations involved in FQ resistance. Genome assemblies of our 2 GBS strains are available in NCBI BioProject (No.: PRJNA607750).

## 3. Results

### 3.1. Clinical Data, Serotyping, and Sequence Types (STs)

The human GBS strain (H21) was isolated from the tissue and pus swab of a 61-year-old male who was hospitalized for an infection of the left foot in 2015 (Table 1). The patient was a repair worker who was not related to fish farms, had no underlying disease, and suffered an injury on the foot. Cultures also grew *Prevotella* species. He was treated with a course of amoxicillin-clavulanate 1 g twice daily orally. The fish GBS strain (F49) was isolated from the flesh of an asymptomatic tilapia in 2016 from a local wet market during our food animal surveillance study.

Multiplex PCR confirmed the serotype of H21 was Ib, whereas the fish GBS F49 belonged to serotype Ia (Table 1). According to whole-genome analysis, H21 belonged to ST651 in clonal complex (CC) 103 (Figure 1), which is more often reported in bovine mastitis [15], whereas the fish GBS F49 was ST7, a sequence type (ST) often reported in streptococcosis outbreaks in freshwater fish [3].

### 3.2. Antimicrobial Susceptibility Testing

Both GBS strains were tested for 13 antibiotics (Table 2). PEN and CTX MICs of 2.0 mg/L were in human GBS strain H21. MDR to ERY, CLI, GEN, CIP, LEV, and TET was also observed. Antimicrobial resistance genes (ARGs) *ermB* (erythromycin), *lnuB (*clindamycin), *tetS (*tetracycline), *CatA8* (chloramphenicol), *ant6-Ia*, *sat4A*, and *aph3-III* (gentamicin) were also found, which corresponded with the resistance phenotypes listed in Table 2.

Fish GBS strain F49 showed a PEN and CTX MIC of 0.5 mg/L and resistance to CIP (4 mg/L), GEN (16 mg/L), and tetracyclines (TET, MIN, and DOX MICs were > 16 mg/L). ARGs conferring resistance to macrolide (*mreA*) and tetracycline (*tetS*) were found in F49.

### 3.3. Novel Amino Acid Substitutions Observed in Penicillin-Binding Proteins and GBS

Sequence alignment of PBPs and fluoroquinolone (FQ)-resistant genes were compared to those in literature [2,7,16,17,18,19,20,21,22]. Non-synonymous amino acid (a.a.) substitutions in PBPs of H21 were found at two sites of PBP1A along with a four-a.a. deletion (_718_NGNG_721_). In addition, three non-synonymous a.a. substitutions were found in PBP2B, one a.a. substitution was found in PBP1B, one a.a. substitution was found in 2A, and two sites were found in PBP2X (Table 3). Of note, T701P substitution in PBP1A has not been reported before, while V726A has just recently been mentioned [21]. Deletion of four a.a.s at _718_NGNG_721_ was present in reported GBS isolates regardless of PEN-NS; thus, it may not be related to antibiotic susceptibility in general [16]. V80A, S147A, and S160A in PBP2B and T720S in PBP2X was mentioned in a GBS cattle study where cephalosporin (CP) and PEN resistance was observed [2]. A.a. substitution in fish GBS F49 was found at only five sites across all PBPs: deletion of _720_NGNG_721_ in PBP1A, E63K in PBP2A, L41S and A95D in PBP1B, and, finally, N192S in PBP2B (Table 3). Among them, PBP2B N192S has not been reported, while the other four mutations were observed in literature [16,17,18,20,21].

### 3.4. Quinolone Resistance and Mutations of Quinolone Resistance Determinant Regions (QRDRs)

Alignment of fluroquinolone (FQ)-resistant determinants, including *gyr*A/B and *par*C/E, were compared between the reference sequences and our GBS isolates. H21 showed unique a.a. substitutions at gyrA (I486V), gyrB (T5I, Q274H, and V498A), and parE proteins (D18N, I148V, and T196I) (Table 4). Apart from V498A in *gyr*B [22], the other six mutation sites were not mentioned in literature. These sites were not within the catalytic site of the QRDR region nor were they reported as mutation hotspots. Sequence alignment of FQ-resistant genes in fish GBS F49 were similar to NEM316; thus, other mechanisms may have caused resistance to ciprofloxacin.

### 3.5. Comparison of Virulence Genes

The virulence genes of H21 and F49 were compared in Table 5. Both strains carried genes encoding adherence, enzymes, and toxins. F49 lacked genes responsible for immune evasion, manganese uptake, and protease production, whereas H21 lacked only the pilus island 1 gene. Notably, H21 carried *sip*, which is a highly conserved immunoreactive antigen that induces cross-protective immunity against GBS infections [23,24].

## 4. Discussion

The firstline drugs for treating human and animal GBS infections are β-lactams. However, PEN-NS GBS has been observed in 0.7% to 6% of overall GBS strains in other countries [17,25,26], while routine susceptibility testing of β-lactams is still not required, as mentioned in CLSI. Although PEN-NS GBS has been reported, PEN and CTX MICs of those strains were all under 1 mg/L. This is the first report in clinical and fish GBS isolates to have observed a PEN and CTX MIC of 2 mg/L, which raises concerns about routine antibiotic susceptibility testing of this species. Surveillance studies have reported PEN MIC of 0.25 mg/L with PBP2X mutations at Q557E or G406D, where the strains belonged to serotype III sequence type (ST) 19 or its single-locus variant (SLV) [17], and G398A from a serotype III ST109 clone (MICs ≥ 0.25 mg/L) [19]. Previously, Hong Kong has reported two GBS PEN-NS strains isolated from patients with bacteremia in 2007 that had PEN MIC 0.19 mg/L and 0.25 mg/L [27], and the latter also revealed an a.a. substitution at Q557E of PBP2X (unpublished). Human GBS H21 (ST651, serotype Ib) showed MDR and a high level of non-susceptibility to PEN and CTX. Infection of GBS ST651 in humans is scarce. To date, only China has reported ST651, which was serotype III, in a pregnant mother from a carriage study [28]. ST651 is a single-locus variant (SLV) of ST103 (Figure 1) and was reported together with another ST103 SLV, ST568, as a predominant strain type causing bovine mastitis in dairy farms of China during 2014–2016 [2]. PEN-NS was observed in all their strains, with 22.5% of isolates showing PEN MIC 2.0 mg/L, ampicillin-NS (90% of strains > 0.25 mg/L), and ceftiofur-NS (≥0.25 mg/L), in which the latter is a third-generation cephalosporin (CP) used in veterinary medicine for metritis and for intra-mammary use in lactating cattle. Incidentally, H21 carried identical a.a. substitutions to the report at PBP2B (V80A, S147A, S160A) and T720S of PBP2X. Based on our a.a. alignments and those described in literature [2,17,18,19], these a.a. substitutions are likely to confer the higher PEN MIC to 2.0 mg/L and CTX-NS. Eburst of CC103 showed 12 SLVs including ST103 (Figure 1). Five of the ten STs had a defined source of bovine origin. Reports of CC103 causing infection and colonization in humans are rare, albeit a recent report suggested the possibility that the bovine-related CC103 can infect humans and tilapias [29,30]. However, this cross-species infection still renders furtherinvestigations. Reduced CP susceptibilities were found among 21 clinical GBS isolates from an Italian study with a.a. substitutions in PBP1A (T145A) and 2X (I377V) [18], the latter of which was present in our human GBS H21. Two new a.a. substitutions, N192S in PBP2B (F49) and PBP1A T701P (H21), were noted in our strains. Many of the functional domains and catalytic sites of the PBPs in GBS have been inferred from the related PBPs of *Streptococcus pneumoniae* and other Streptococci. The significance of these two a.a. substitutions of PBP1A (T701P) and PBP2B (N192S) on PEN and CP resistance in our strains remains elusive. The a.a. substitution of N192S in PBP2B is near to the N-terminus of the protein and outside of the transpeptidase domain, as aligned with the a.a. sequence of PBP2B of *Streptococcus pneumoniae* R6 (Genbank accession no.: NC_003098.1), which has the highest similarity in a.a. sequence identity to our protein [31]. The T701P a.a. substitution is located close to the C-terminus and away from the PBP catalytic sites of PBP1a of, namely, _370_SXXK_373_, _428_SXN_430_, and _557_KTG_559_ of the corresponding PBP1a of *S. pneumoniae* [32]. Interestingly, a further search with blast (Genbank) of this a.a. sequence retrieved four further strains with 100% homology (accession no.: AKI56779.1, WP_017647107.1), one of which was from that of a whole-genome sequence of GBS of ST103 [33], the founder ST associated with strains from bovine mastitis. However, no information was available regarding its PEN and CP susceptibilities. Laboratory work with mutagenesis of these newly described point mutations should be further elucidated to understand their role in PEN and CP resistance. Lack of protein prediction analysis and functional assays of the PBP substitutions to pinpoint the relation of these mutations to raised PEN MIC is also a limitation of our study, especially in PBP2B, where the mutations were at two serine sites, a common amino acid found in catalytic sites.

FQ resistance observed in H21 was 64-fold above the breakpoint. FQ resistance has been reported mainly in STs 1, 10, and 19 in other countries, but not ST651 [7,22,34]. A.a. substitutions at gyrB (T5I, Q274H) and parE (D18N, I148V, I196I) were not previously reported. The relation of these mutations to FQ resistance remains to be elucidated. Quinolones are not used for treatment in GBS infections, but resistance to these drugs has been increasingly observed; thus, FQ resistance maybe included into surveillance in the future.

## 5. Conclusions

MDR and PEN-NS GBS is emerging. This limits our choice of treatment for GBS infections in all patient types, including pregnant mothers and neonates and as well as in GBS-related animal diseases. Thus, a revisit of CLSI guideline for testing of PEN MICs and PEN meningitis breakpoints with reference to *Streptococcus pneumoniae* should be considered, especially as these strains are increasingly identified from patients with sepsis and meningitis. Increased antimicrobial resistance in animal GBS strains calls for deep concern about farmed fish and animal products and how these could potentially interrelate to affect human health and infection control. Finally, increased reports on FQ resistance in GBS suggest the inclusion of FQ in surveillance studies should be considered.

## Figures and Tables

**Figure 1 microorganisms-08-01055-f001:**
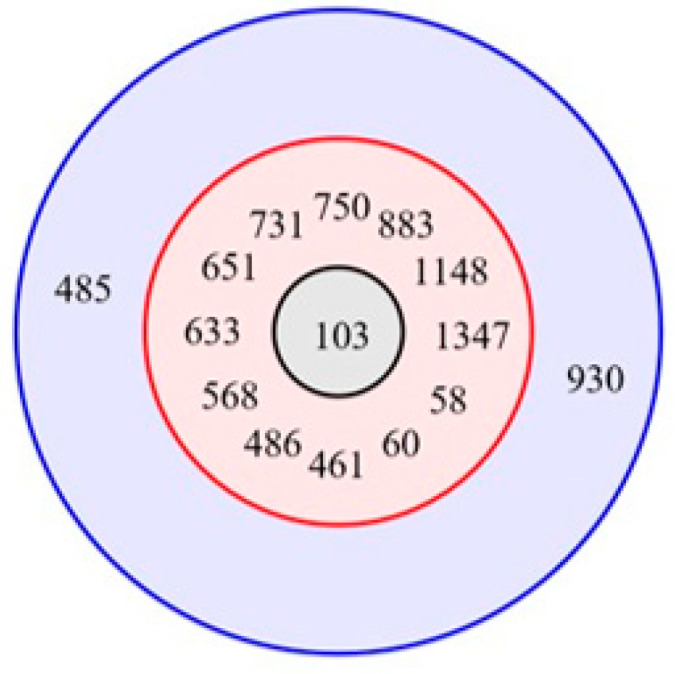
Eburst diagram of sequence types in clonal cluster 103. The diagram shows the single-locus variants (red ring) and the double-locus variants (blue ring) of sequence type (ST) 103 (grey circle in the center) in clonal cluster 103.

**Table 1 microorganisms-08-01055-t001:** Characteristics of Group B *Streptococcus* (GBS) isolates.

Isolate	H21	F49
Host	Human	Fish
Year of isolation	2015	2016
Gender	Male	NA
Age (yr)	61	NA
Host health status	Left foot infection	Healthy
Specimen site	Wound	Flesh
Serotype	Ib	Ia
Sequence type (ST)	651	7
Clonal cluster (CC)	103	7

**Table 2 microorganisms-08-01055-t002:** Minimum inhibition concentrations (MICs) of GBS isolates to antibiotics.

Isolate	MIC (mg/L)		MIC Breakpoints (mg/L)		
Antibiotics	H21	F49	Sensitive	Intermediate	Resistant
Penicillin (PEN) *	2	0.5	≤0.12	-	-
Cefotaxime (CTX) ^#^	2	0.5	≤0.5	-	-
Vancomycin (VAN)	1	0.5	≤1	-	-
Erythromycin (ERY)	**>16**	≤0.12	≤0.25	0.5	≥1
Clindamycin (CLI)	**>16**	≤0.12	≤0.25	0.5	≥1
Gentamicin (GEN)	32	16	≤1	-	-
Ciprofloxacin (CIP) ^^^	**32**	**4**	≤1	2	≥4
Levofloxacin (LEV)	**32**	1	≤2	4	≥8
Tetracycline (TET)	**16**	**>16**	≤2	4	≥8
Minocycline (MIN)	≤0.12	**>16**	≤2	4	≥8
Doxycycline (DOX)	1	**>16**	≤2	4	≥8
Linezolid (LNZ)	1	1	≤2	-	-
Chloramphenicol (CHL)	**8**	≤0.12	≤4	8	≥16
Inducible Clindamycin Resistance	NA	Neg			

MICs that are non-susceptible are underlined. MICs that are intermediate (namely, F49 ciprofloxacin (CIP) and H21 chloramphenicol (CHL)) or resistant are in bold font. NA, not applicable. MIC breakpoints for GBS were referenced by CLSI [9]. * Penicillin non-susceptible (PEN-NS) strains were confirmed by E-test, which also showed 2.0 mg/L and 0.38 mg/L for H21 and F49, respectively. ^#^ E-test confirmed cefotaxime non-susceptibility of H21 and F49 with MICs of 2.0 mg/L and 0.5 mg/L, respectively. ^^^ CLSI breakpoint for *Enterococcus* spp. [9].

**Table 3 microorganisms-08-01055-t003:** Amino acid substitutions of penicillin-binding proteins PBP1A, 1B, 2A, 2B, and 2X.

**Amino Acid Substitutions Identified in PBPs by Amino Acid Position**															
**Strain Name**	**PBP1A**														
	**PEN MIC (mg/L)**	**701 ***	**718**	**719**	**720**	**721**	**722**	**723**	**724**	**725**	**726**	**727**	**728**	**729**	**730**
2603V/R	0.06 ^#^	T	N	G	N	G	N	N	N	T	V	P	N	G	N
NEM316	0.06 ^#^	.	.	.	--	--	.	.	.	.	.	.	.	.	.
H21	2.0	P	--	--	--	--	.	.	.	.	A	.	.	.	.
F49	0.5	.	.	.	--	--	.	.	.	.	.	.	.	.	.
NY1512 ^a^	2.0														
SMX1626 ^a^	2.0														
SQ1615 ^a^	2.0														
NY1547 ^a^	1.0														
**Amino Acid Substitutions Identified in PBPs by Amino Acid Position**															
**Strain Name**	**PEN MIC (mg/L)**	**PBP1B**			**PBP2A**		**PBP2B**					**PBP2X**			
		**41**	**95**		**63**		**80**	**147**	**160**	**192 ***		**336**	**377**	**425**	**720**
2603V/R	0.06 ^#^	L	A		E		V	S	S	N		Y	I	K	T
NEM316	0.06 ^#^	.	D		.		.	.	.	.		.	V	.	.
H21	2.0	.	D		K		A	A	A	.		.	V	.	S
F49	0.5	S	D		K		.	.	.	S		.	.	.	.
NY1512 ^a^	2.0	.	D				A	A	A	.		F	V	M	S
SMX1626 ^a^	2.0	.	D				A	A	A	.					
SQ1615 ^a^	2.0	.	D									F	V	M	S
NY1547 ^a^	1.0						A	A	A	.					

Sequences were aligned to the corresponding amino acid (a.a.) sequences of 2603V/R and NEM316 (accession numbers NC_004116.1 and NC_004368). Period (.) indicates identical a.a.; dash (--) indicates deletion of a.a. leading to truncation. ^#^ MIC values adopted from Nagano et al. 2008 [16]. * PBP mutations that were not previously reported. ^a^ Sequences available from Genbank according to Hu et al., 2018 [2]. Accession numbers of NY1512 PBP1B, PBP2B, and PBP2X genes were KX374357, KX374358, and KX374359, respectively; SMX1626 PBP1B and 2B genes were KX374364 and KX374365, respectively; SQ1615 PBP1B and 2X genes were KX374368 and KX374369, respectively; and NY1547 PBP2B gene was KX374361. a.a. sequences not available for analysis were left blank.

**Table 4 microorganisms-08-01055-t004:** Amino acid substitutions of quinolone-resistant determinants gyrA/B and parC/E.

**Amino Acid Substitutions Identified in GyrA/B Proteins by Amino Acid Position**											
**Strain Name**	**MIC (mg/L)**		**GyrA**				**GyrB**				
	**CIP**	**LEV**	**307**	**371**	**486 ***		**5 ***	**274 ***	**498**		
2603V/R	-	0.5 ^#^	A	G	I		T	Q	V		
NEM316	-	0.5 ^#^	D	E	.		.	.	.		
H21	32	32	D	E	V		I	H	A		
F49	4	1	.	.	.		.	.	.		
**Amino Acid Substitutions Identified in ParC/E Proteins by Amino Acid Position**											
**Strain Name**	**MIC (mg/L)**		**ParC**				**ParE**				
	**CIP**	**LEV**	**639**	**640**	**641**		**18 ***	**148 ***	**196 ***	**499**	**507**
2603V/R	-	0.5 ^#^	S	V	E		D	I	T	L	I
NEM316	-	0.5 ^#^	N	.	D		.	.	.	I	V
H21	32	32	.	.	.		N	V	I	.	V
F49	4	1	N	.	D		.	.	.	I	V

Sequences were aligned to the corresponding a.a. sequences of 2603V/R and NEM316 (accession numbers NC_004116.1 and NC_004368). ^#^ MIC values adopted from Nagano et al. 2008 [16]. Period (.) indicates identical a.a. * Mutations that were not previously reported.

**Table 5 microorganisms-08-01055-t005:** Virulence genes comparison of H21 and F49.

	Strain ID	H21	F49
	Source	Human	Tilapia
	Serotype	Ib	Ia
	ST	651	7
Adherence	*pavA*	●	○
	*fbsA*	●	○
	*PS-1*	○	●
	*pilB*	●	○
	*plr/gapA*	●	○
Enzymes	*hylB*	●	●
	*eno*	●	○
Immunoreactive antigen	*sip*	●	○
Manganese uptake	*psaA*	●	○
Protease	*cppA*	●	○
	*scpA/scpB*	●	○
	*htrA/degP*	●	○
Toxins	*cylX*	●	●
	*cylD*	●	●
	*cylG*	●	●
	*acpC*	●	●
	*cylZ*	●	●
	*cylA*	●	●
	*cylB*	●	●
	*cylE*	●	○
	*cylF*	●	●
	*cylI*	●	●
	*cylJ*	●	●
	*cylK*	●	●

○—Indicates the absence of gene. ●—Indicates the presence of gene.

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
