# Peer review of "Multidrug-Resistant Streptococcus agalactiae Strains Found in Human and Fish with High Penicillin and Cefotaxime Non-Susceptibilities"

_microorganisms, 2020, doi:10.3390/microorganisms8071055_

Round 1
Reviewer 1 Report
Manuscript details:
Journal: Microorganisms
Manuscript ID: microorganisms-849507
Type of manuscript: Communication
Title: Multidrug resistant Streptococcus agalactiae strains found in human
and fish with high penicillin and cefotaxime non-susceptibilities
Authors: Carmen Li, Dulmini Nanayakkara Sapugahawatte, Ying Yang, Kam Tak
Wong, Norman Wai Sing Lo, Margaret Ip *
Dr. C Li et al analyzed 2 isolates of GBS with high PCG and CTX MIC values. One is clinical isolates and another is an isolate from fish. They performed NGS analysis and found amino acid substitutions in PBPs and GyrA/B and ParC/E.
As far as I know, this is the first case of PRGBS with confirmed amino acid substitutions in PBPs of clinical isolate in China. Moreover, this is also the first case of PRGBS isolate from fish. Therefore, this manuscript contains valuable information concerning PRGBS.
However, before my reading this manuscript, I did not fully believe the contents in Ref. 2 (Hu Y et al.). Because MICs of PCG for all isolates in Ref. 2 were relatively high, I could not deny the possibility of the measurement error in MIC determination. Moreover, because most amino acid substitutions in PBPs are not previously reported and there was not the evidence using allelic exchanged strains for elevated MICs in Ref.2, I did not fully believe the contents in Ref. 2 (Hu Y et al.). However, authors reported identical amino acid substitutions in PBPs independently of the group of Ref. 2. Therefore, I think that these amino acid substitutions in PBPs may contribute to reduced beta-lactam susceptibility. However, lack of the evidence using allelic exchanged strains for elevated MICs is a limitation of this study. Therefore, authors should write the limitation of this study in the discussion section.
Minor comment
- Page 2 Line56: The number of clinical isolates is accurate? 3000 isolates? Correct number is needed.
Author Response
Reviewer: 1 (Comments to the Author)
Dr. C Li et al analyzed 2 isolates of GBS with high PCG and CTX MIC values. One is clinical isolates and another is an isolate from fish. They performed NGS analysis and found amino acid substitutions in PBPs and GyrA/B and ParC/E.
As far as I know, this is the first case of PRGBS with confirmed amino acid substitutions in PBPs of clinical isolate in China. Moreover, this is also the first case of PRGBS isolate from fish. Therefore, this manuscript contains valuable information concerning PRGBS.
However, before my reading this manuscript, I did not fully believe the contents in Ref. 2 (Hu Y et al.). Because MICs of PCG for all isolates in Ref. 2 were relatively high, I could not deny the possibility of the measurement error in MIC determination. Moreover, because most amino acid substitutions in PBPs are not previously reported and there was not the evidence using allelic exchanged strains for elevated MICs in Ref.2, I did not fully believe the contents in Ref. 2 (Hu Y et al.). However, authors reported identical amino acid substitutions in PBPs independently of the group of Ref. 2. Therefore, I think that these amino acid substitutions in PBPs may contribute to reduced beta-lactam susceptibility. However, lack of the evidence using allelic exchanged strains for elevated MICs is a limitation of this study. Therefore, authors should write the limitation of this study in the discussion section.
Response:
Thank you. We concur with the reviewer that the lack of experimentation on the allelic exchange of these proteins limits the association of the mutations with the elevated MICs. The limitation of the study is now mentioned in lines 224-226.
Minor comment
- Page 2 Line56: The number of clinical isolates is accurate? 3000 isolates? The correct number is needed.
Response:
These isolates include those from pregnant mothers for the screening of GBS carriage from rectovaginal swabs, as well as other admissions with GBS disease during the period 2014-2017. Among them, 2517 single patient isolates were archived in our laboratory for clinical surveillance, where the PEN-NS strain was picked up. The number has been adjusted in lines 56-57.
Reviewer 2 Report
This is a report of 2 Streptococcus agalactiae (GBS) strains, isolated from human wound and fish, with high penicillin and cefotaxime non-susceptibilities. GBS non-susceptibility to penicillin is not common with only a handful of publications published in the past. The manuscript is written well and I would like to suggest just a couple of points.
Suggestions,
1, Result 3.1
H21 belongs to ST651 in Clonal Complex (CC) 103 (Figure 1)
Please add some information and references about the CC103 since there is no explanation in the introduction or the result section. Also please add some explanation in the figure legend to make the point of this figure clear.
2, Result 3.2
Please indicate the range of MIC value for Resistant, non-susceptible, and susceptible which will help understand the numbers better.
3, Result 3.3
Line 129 “(Error! Reference 129 ource not found.)” Please correct.
4, Discussion
Line 166-167; PEN-NS GBS has been observed in 15% of overall GBS strains in other countries,
Please add references for this description.
Author Response
Reviewer: 2 (Comments to the Author)
This is a report of 2 Streptococcus agalactiae (GBS) strains, isolated from human wound and fish, with high penicillin and cefotaxime non-susceptibilities. GBS non-susceptibility to penicillin is not common with only a handful of publications published in the past. The manuscript is written well and I would like to suggest just a couple of points.
Suggestions,
1, Result 3.1
H21 belongs to ST651 in Clonal Complex (CC) 103 (Figure 1)
Please add some information and references about the CC103 since there is no explanation in the introduction or the result section. Also please add some explanation in the figure legend to make the point of this figure clear.
Response:
CC103 has been documented from strains in bovine mastitisa,b. GBS infection in humans from CC103 is rare though human infection of ST485 (double locus variant of ST103 in CC103) has just been reported to be prevalent in Chinac. The explanation has been added in the Discussion section from lines 205 to 207, and in Figure 1.
References:
- Carvalho-Castro, G. A.; Silva, J. R.; Paiva, L. V.; Custodio, D. A. C.; Moreira, R. O.; Mian, G. F.; Prado, I. A.; Chalfun-Junior, A.; Costa. G. M. Molecular epidemiology of Streptococcus agalactiae isolated from mastitis in Brazilian dairy herds. 2017. 48, 3, 551-559.
- Reyes, L.; Chaffer, M.; Rodrigues-Lecompte, J. C.; Sanchex, J.; Zadoks, R. N.; Robinson, N.; Cardona, X.; Ramirex, N.; Keefe, G. P. Short communication: Molecular epidemiology of Streptococcus agalactiae differs between countries. 2017. J Dairy Science, 100, 11, 9294-9297 doi: 10.3168/jds.2017-13363
- Ran, R.; Li, LP.; Huang, T.; Huang, Y.; Huang, W.; Yang, X.; Lei, A.; Chen, M. Phylogenetic, comparative genome and structural analyses of human Streptococcus agalactiae ST485 in China. 2018, 19, 716. Doi 10.1186/s123864-018-5084-0
2, Result 3.2
Please indicate the range of MIC value for Resistant, non-susceptible, and susceptible which will help understand the numbers better.
Response:
In order to fit in the MIC ranges of resistant, non-susceptible, and susceptible to the antibiotics, Table 1 has been separated into 2 tables: Table 1 now shows the characteristics of GBS isolates, while Table 2 shows the MIC and the breakpoints according to CLSI (2018). The referring Tables in-text has also been modified (lines 100, 106 for Table 1; lines 114 and 118 for Table 2).
3, Result 3.3
Line 129 “(Error! Reference 129 ource not found.)” Please correct.
Response:
The error of the linked table has been corrected, currently in line 144. We apologize for our carelessness.
4, Discussion
Line 166-167; PEN-NS GBS has been observed in 15% of overall GBS strains in other countries,
Please add references for this description.
Response:
The wordings for the sentence have been changed to a more accurate form and references have been added in lines 182-184.
Reviewer 3 Report
Author (s) have recently identified two GBS strains with PEN-NS from a patient (H21) and from a tilapia fish (F49), do characterize these 2 GBS strains in view of its antibiotic resistance, found novel amino acid substitutions observed in penicillin-binding proteins and of GBS in this study and it is important finding and below is my question to author (s) for publication.
Major revision
1. Author (s) mentioned that T701P substitution in PBP1A has not been reported and PBP2B N192S has not also been reported.
Could author (s) show some data (or more detail explain) about the contribution (or role) of antibiotics resistances of this substitution in result or discussion?
Minor revision
- In line 127, Please write abbreviation about cephalosporin and penicillin.
- In line 129-130, Could you correct about “Error! Reference ource not found”?
- In line 180, Could you correct “2” in 2014-20162?
- In line 188, Please write abbreviation about cephalosporins.
- In line 195, Please write abbreviation about Quinolones.
- In line 160, Please omit “gene” next “sip”.
Author Response
Author (s) have recently identified two GBS strains with PEN-NS from a patient (H21) and from a tilapia fish (F49), do characterize these 2 GBS strains in view of its antibiotic resistance, found novel amino acid substitutions observed in penicillin-binding proteins and of GBS in this study and it is important finding and below is my question to author (s) for publication.
Major revision
- Author (s) mentioned that T701P substitution in PBP1A has not been reported and PBP2B N192S has not also been reported.
Could author (s) show some data (or more detail explain) about the contribution (or role) of antibiotics resistances of this substitution in result or discussion?
Response:
Much of the functional domains and catalytic sites of the PBPs in GBS have been inferred from the related PBPs of Streptococcus pneumoniae and other Streptococci. It remains elusive of the significance of these two a.a. substitutions of PBP1A (T701P) and PBP2B (N192S) on PEN and CP resistance in our strains. The a.a. substitution of N192S in PBP2B is near to the N-terminus of the protein and outside of the transpeptidase domain as aligned with the a.a. sequence of PBP2B of Streptococcus pneumoniae R6 (Genbank accession no.: NC_003098.1) which has the highest a.a. sequence identity to our protein [a]. The T701P a.a. substitution is located close to the C-terminus, and away from the PBP catalytic sites of PBP1a of namely, 370SXXK373, 428SXN430 and 557KTG559 of the corresponding PBP1a of S. pneumoniae [b]. Interestingly, further search with blast on Genbank of this a.a. sequence retrieved four further strains with 100% homology (Accession No: AKI56779.1, WP_017647107.1), one of which was from that of a whole genome sequence of GBS of ST103 [c], the founder ST associated with strains from bovine mastitis. However, no information was available regarding its PEN and CP susceptibilities. Laboratory work with mutagenesis of these newly described point mutations should be further elucidated to understand their role in PEN and CP resistance. This information has been added to the discussion section at line 216-224
- Ramalingam, J.; Vennila, J.; Subbiah, P. Computational studies on the resistance of penicillin-binding protein 2B (PBP2B) of wild-type and mutant strains of Streptococcus pneumoniae against β-lactam antibiotics. Chem Biol Drug Des 2013, 82, 275-289, DOI 10.1111/j.1747-0285.2012.01387.x.
- Job, V.; Carapito, R.; Vernet, T.; Dessen, A.; Zapun, A. Common alterations in PBP1a from resistant Streptococcus pneumoniae decrease its reactivity toward beta-lactams: structural insights. J Biol Chem 2008, 283, 4886-4894, DOI M706181200.
- de Aguiar, E.L.; Mariano, D.C.; Viana, M.V.; Benevides Lde, J.; de Souza Rocha, F.; de Castro Oliveira, L.; Pereira, F.L.; Dorella, F.A.; Leal, C.A.; de Carvalho, A.F.; Santos, G.S.; Mattos-Guaraldi, A.L.; Nagao, P.E.; de Castro Soares, S.; Hassan, S.S.; Pinto, A.C.; Figueiredo, H.C.; Azevedo, V. Complete genome sequence of Streptococcus agalactiae strain GBS85147 serotype of type Ia isolated from human oropharynx. Stand Genomic Sci 2016, 11, 39-6. eCollection 2016, DOI 10.1186/s40793-016-0158-6.
Minor revision
In line 127, Please write abbreviation about cephalosporin and penicillin.
Response:
We have now added the abbreviations to cephalosporin (CP) and penicillin (PEN) (line 142).
In line 129-130, Could you correct about “Error! Reference ource not found”?
Response:
Please refer to Question 3 of Reviewer 2.
In line 180, Could you correct “2” in 2014-20162?
Response:
The correction has been made (line 197).
In line 188, Please write abbreviation about cephalosporins.
Response:
The abbreviation for cephalosporins (CP) is now included in the manuscript.
In line 195, Please write abbreviation about Quinolones.
Response:
As the word “quinolones” only appeared once, we think there is no need for an abbreviation (line 231).
In line 160, Please omit “gene” next “sip”.
Response:
The correction has been made (line 175).
Reviewer 4 Report
This manuscript describes a penicillin nonsusceptible clinical strain of Streptococcus agalactiae with MIC of 2 mg. This study has some meaningful data for the emergence of antibiotic resistant strain of S. agalactiae in human.
Some suggestions are written in below.
- The authors should describe the selection criteria of these 2 isolates from 3,000 or 126 specimens. It is necessary to emphasize the meaning of this study.
- MIC and Etest were performed. Please show both of the microdilution MIC results and Etest results.
- Table 2. Check that two or three sites of PBP1A in the manuscript.
- The methods for the assembly of NGS data should be mentioned.
- Please describe the method to find the vir genes in Table 4.
- HGBS or FGBS seem to be confusing. Human or fish would be good.
Author Response
This manuscript describes a penicillin nonsusceptible clinical strain of Streptococcus agalactiae with MIC of 2 mg. This study has some meaningful data for the emergence of antibiotic resistant strain of S. agalactiae in human.
Some suggestions are written in below.
- The authors should describe the selection criteria of these 2 isolates from 3,000 or 126 specimens. It is necessary to emphasize the meaning of this study.
Response:
Please refer to the answer to Question 1 of Reviewer 1.
- MIC and Etest were performed. Please show both of the microdilution MIC results and Etest results.
Response:
Etest results are noted below Table 2.
- Table 2. Check that two or three sites of PBP1A in the manuscript.
Response:
There were a total of 3 alterations in PBP1A: 2 mutations (at T701P and V726A) and a 4 amino acid deletion (718 to 721). More accurate wordings in describing PBP1A and PBP2B alterations are now used (lines 135-138).
- The methods for the assembly of NGS data should be mentioned.
Response:
The methods of NGS assembly have been described with more detail in section 2.5. Whole genome sequencing (line 80-96).
- Please describe the method to find the vir genes in Table 4.
Response:
Methods for finding the virulence genes have been added (line 87 to line 89).
- HGBS or FGBS seem to be confusing. Human or fish would be good.
Response:
This has been changed accordingly throughout the manuscript.

Round 2
Reviewer 3 Report
Nothing